# The Current Role of Peptide Receptor Radionuclide Therapy in Meningiomas

**DOI:** 10.3390/jcm11092364

**Published:** 2022-04-23

**Authors:** Christina-Katharina Fodi, Jens Schittenhelm, Jürgen Honegger, Salvador Guillermo Castaneda-Vega, Felix Behling

**Affiliations:** 1Department of Neurosurgery and Neurotechnology, University Hospital Tübingen, Eberhard-Karls University, 72076 Tübingen, Germany; christina-katharina.fodi@med.uni-tuebingen.de (C.-K.F.); juergen.honegger@med.uni-tuebingen.de (J.H.); 2Center for CNS Tumors, Comprehensive Cancer Center Tübingen-Stuttgart, University Hospital Tübingen, Eberhard-Karls-University, 72076 Tübingen, Germany; jens.schittenhelm@med.uni-tuebingen.de; 3Department of Neuropathology, University Hospital Tübingen, Eberhard-Karls University, 72076 Tübingen, Germany; 4Department of Nuclear Medicine and Clinical Molecular Imaging, University Hospital Tübingen, Eberhard-Karls University, 72076 Tübingen, Germany; salvador.castaneda@med.uni-tuebingen.de; 5Werner Siemens Imaging Center, Department of Preclinical Imaging and Radiopharmacy, Eberhard-Karls University, 72076 Tübingen, Germany

**Keywords:** meningioma, peptide receptor radionuclide therapy, PRRT, targeted therapy, somatostatin receptor, SSTR

## Abstract

Meningiomas are the most common primary intracranial tumors. The majority of patients can be cured by surgery, or tumor growth can be stabilized by radiation. However, the management of recurrent and more aggressive tumors remains difficult because no established alternative treatment options exist. Therefore, innovative therapeutic approaches are needed. Studies have shown that meningiomas express somatostatin receptors. It is well known from treating neuroendocrine tumors that peptide radioreceptor therapy that targets somatostatin receptors can be effective. As yet, this therapy has been used for treating meningiomas only within individual curative trials. However, small case series and studies have demonstrated stabilization of the disease. Therefore, we see potential for optimizing this therapeutic option through the development of new substances and specific adaptations to the different meningioma subtypes. The current review provides an overview of this topic.

## 1. Introduction

With an incidence of 8.81/100,000, meningiomas are the most common primary intracranial tumor [1]. These slow-growing tumors develop along the meningeal coverings of the cerebral convexities, skull base, and spine, and in rare cases even within the ventricles. Depending on the size and location, a variety of different symptoms and syndromes can emerge [2]. Treatment is advisable for lesions that show concerning growth, cause symptoms, or have a significant size in a critical location. Most meningiomas can be effectively treated by surgical resection. Radiation therapy can be applied in selected cases but is particularly important in recurrent cases or for meningiomas that are histopathologically graded as more aggressive [3]. Other than surgery and radiation, there are no established treatment modalities outside of previous clinical trials [4]. The management of meningiomas that recur after resection and radiation treatment remains especially challenging, and alternative treatment options are urgently needed. Targeted therapies based on molecular aberrations [5] are currently under investigation in several clinical trials, with a focus on high-grade and recurrent meningiomas (e.g., NCT02648997, NCT03279692, and NCT03631953). However, an innovative treatment option for recurrent meningiomas has been applied in difficult cases. Peptide receptor radionuclide therapy (PRRT) utilizes the expression of somatostatin receptors (SSTRs) in meningioma tissue. Somatostatin analogs coupled to a radionuclide apply radiation directly to tissue that expresses somatostatin receptor 2A, such as meningioma cells [6,7]. In recent years, this targeted treatment approach has been investigated for use against meningiomas. This review focuses on the current use of PRRT to treat meningiomas and provides an outlook regarding its potential future therapeutic role.

## 2. Peptide Receptor Radionuclide Therapy

### 2.1. Somatostatin Receptors

Somatostatin receptors, through which somatostatin exerts its effects, were first described in 1978 by Schonbrunn and Tashjian [8]. Five different subtypes have been discovered in humans and other mammals. The SSTR2 subtype has one intron, which can result in two different receptors, SSTR2A and the SSTR2B, by alternative splicing. The remaining subtypes do not possess introns [9]. However, SSTR2B has only been detected in mouse tissue, and it is unclear whether this subtype is also expressed in humans [10]. These G protein-coupled receptors are composed of glycoproteins [11] and are membrane-bound with seven alpha-helical transmembrane domains [10]. They have an extracellular N-terminal end, which is responsible for binding the specific ligand somatostatin. The C-terminal end is intracellular and transmits signal transduction [10] through a heterotrimeric G protein that consists of an α-, β-, and γ-subunit and triggers different intracellular pathways with the help of GTP. Upon binding of a ligand to the SSTR, the cascade further triggers activation or inhibition of cytoplasmic targets via membrane-bound proteins [12]. Examples of commonly triggered pathways include the adenylate cyclase and the phospholipase C system, which act as signal amplifiers. However, the direct stimulation of potassium or calcium channels can also be triggered to achieve the desired effects [13,14]. Somatostatin is involved in numerous physiological regulatory mechanisms in several different organ systems, including the inhibition of endocrine and exocrine secretions, gastrointestinal motility and nutrient absorption, and neurotransmitter regulation [15].

A central task of SSTR is the inhibition of cell proliferation. The antiproliferative effects of somatostatin and its receptors are mediated through several mechanisms. SSTR2 and SSTR3 can induce apoptosis of single cells via p53 or independently of p53. SSTR1, SSTR4, and SSTR5 can induce cell arrest in the G1 phase of the cell cycle via the modulation of mitogen-activated protein (MAP) kinase [16]. There are numerous other modes of action of SSTRs that are reserved for their respective tissues. SSTRs occur in varying densities and in different expression patterns. They are found in the brain, pituitary gland, and peripheral nervous system. In the gastrointestinal tract, they are found in large numbers in the stomach, duodenum, jejunum, and pancreas. They are also present in the kidneys, adrenal glands, thyroid gland, and immune cells [11,17,18].

An important discovery was the expression of SSTRs in a large variety of tumors, including most neuroendocrine tumors, such as gastroenteropancreatic neuroendocrine tumors (GEP-NETs) and carcinoids. SSTRs also occur in renal cell carcinoma, breast cancer, and lymphoma. Furthermore, SSTRs are expressed in brain tumors, such as glial tumors, pituitary adenomas, and meningiomas [18,19]. Studies conducted on meningiomas have shown the presence of all five different SSTR subtypes in varying degrees of expression. SSTRs could be detected by immunohistochemistry and by mRNA identification with RT-PCR [20,21,22,23,24]. In these studies, SSTR2A showed the strongest expression [20,21,22,23,25]. We performed a large retrospective analysis of immunohistochemical expression of all five SSTRs in 726 meningiomas. This large cohort also showed strong expression of SSTR1 and SSTR5 and different expression patterns within various clinical subgroups, such as neurofibromatosis type 2 and WHO grades II and III meningiomas [24].

Whether somatostatin expression is linked to tumor proliferation or progression remains unclear, especially because its main function is essentially antiproliferative.

### 2.2. History and Development

In 1987, Krenning et al. examined 1000 patients with various tumors by octreotide receptor scintigraphy (Octreoscan) to visualize SSTR expression. It was hypothesized that tumors showing high uptake may respond well to therapy with somatostatin analogs [26] because somatostatin exerts an antiproliferative effect. Furthermore, the idea of combining somatostatin with radiation was developed so that the radiation dose could be directly delivered to the tumor tissue. These so-called “theranostic substances”, with diagnostic (peptide receptor scintigraphy) and therapeutic (peptide receptor radionuclide therapy) features, were first used in neuroendocrine tumors [27]. The first instance of PRRT was in the early 1990s for a patient with metastatic glucagonoma, which resulted in tumor growth impairment as well as decreasing levels of circulating glucagon [28].

Initially, 111In-DTPA octreotide, which emits Auger and conversion electrons, was applied for PRRT, with SSTR as the target. However, this compound exhibited affinity exclusively for SSTR2 and was additionally not suitable for commercially available ß-emitters such as ^90^Y and ^77^Lu. Therefore, other somatostatin analogs such as DOTATOC and DOTATATE were coupled with the corresponding ß-emitters to form ^177^Lu-DOTATATE and ^90^Y-DOTATOC [29] and used in subsequent studies. PRRT was increasingly used in the treatment of neuroendocrine tumors. It is often applied for metastasized tumors for which the usual treatment methods, such as surgery or localized radiation therapy, are no longer suitable options. Often, standard chemotherapy is no longer sufficient or does not achieve adequate symptom control. In these cases, treatment can be carried out with PRRT [29,30,31]. The phase III NETTER-1 trial demonstrated that PRRT significantly improved the quality of life and progression-free survival of patients with midgut neuroendocrine tumors who received ^177^Lu-DOTATATE and high-dose octreotide [32]. With the knowledge that several tumor types highly express SSTRs [33], PRRT has also been used therapeutically for other tumor types as an individual curative attempt after the exhaustion of current treatment methods [34,35,36].

In many cases, meningiomas can be cured by surgical excision or stabilized radiation therapy [3]. However, sometimes these treatments are not sufficient, and it becomes difficult to treat recurrences or multifocal occurrences of meningiomas; for example, a meningiomatosis cerebri. This often affects patients suffering from neurofibromatosis type 2 (NF2) or tumors corresponding to WHO grades II or III. Through various studies, we know that virtually all meningiomas express SSTRs, although to a varying extent [20,22,24,25,37]. This has been recognized as an opportunity to apply PRRT in individual cases of advanced meningioma [29,38].

### 2.3. Clinical Application and Experience in Meningioma

Certain conditions must be met to perform PRRT. To assess whether there is sufficient receptor expression, the standard procedure prior to therapy is to detect it via peptide receptor positron emission tomography (PET), e.g., with ^68^Ga. This also includes imaging the kinetics of the therapeutic substance [39]. Once sufficient receptor expression has been determined through PET imaging, the patient becomes a candidate for therapy. Somatostatin analogs, such as DOTATOC or DOTATATE, which bind to SSTRs (particularly SSTR2A), are administered intravenously. The administered peptides are coupled to a ß-emitter, usually ^90^Yttrium (Y). This coupling allows the systemic delivery of a cytotoxic level of radiation to individual target cells expressing SSTRs [40]. In addition, adjacent tumor cells that do not necessarily express SSTRs are also irradiated. Alternatively, small studies and case reports have shown that intra-arterial administration of the radiopharmaceutical increases the uptake in meningiomas [41,42,43].

However, there are no specific guidelines regarding this treatment for meningiomas, but general standards for PRRT in NETS are applied. A glomerular filtration rate of at least 40 mL/min or a creatinine value <2.0 mg/dL must be present in order to assure sufficient clearance and to maintain functional organ reserve because the treatment is nephrotoxic. Likewise, there is a risk of hematotoxicity and pretherapeutic cutoffs for platelet and leukocyte count and hemoglobin need to be considered. Therefore, regular monitoring is necessary, which may lead to a pause in treatment [44]. There could also be a risk of acute bone marrow toxicity, particularly in patients who received extended external beam radiation or myelotoxic chemotherapy before PRRT. The number of cycles for non-compromised patients varies from two to five for ^90^Y-DOTATATE/-DOTATOC and ^177^Lu-DOTATATE/-DOTATOC. The time interval between cycles ranges between 6 and 12 weeks. The approved activity levels range from 2.78–4.44 GBq for ^90^Y-DOTATATE/-DOTATOC and from 5.55–7.4 GBq for ^177^Lu-DOTATATE/-DOTATOC. Therapy needs to be adjusted for patients with restricted renal function or borderline bone marrow capacity [45].

Currently, there are only a few studies on PRRT for meningiomas with small patient cohorts. However, some have shown stabilization of the disease. In one study, 5–15 GBq of ^90^Y-DOTATOC was administered to a group of 29 patients with recurrent meningiomas, and disease stabilization was observed in 66% of the patients [6]. Another study demonstrated that a combination of external beam radiation therapy (EBRT) and PRRT was well-tolerated in patients with unresectable primary or recurrent meningioma and resulted in disease stabilization in 7 of the 10 patients [46]. These results confirmed the findings of a previous analysis of the outcome of patients treated with EBRT and PRRT in combination. In that study, ten patients with unresectable WHO grades I or II meningiomas experienced disease stabilization [47]. PRRT was also investigated in patients suffering from NF2. They received four cycles of ^177^Lu-DOTATOC with a median activity of 7.4 GBq. Tumor stabilization occurred in six of eleven patients [48]. A meta-analysis regarding PRRT in meningiomas was performed by Miriam et al. They included 111 patients with treatment-refractory meningiomas, and 63% achieved disease control. The 6-month progression-free survival rates and the 1-year overall survival decreased with higher WHO grades. Explicit guidelines for the treatment of meningioma using SSTR-targeted PRRT have still yet to be internationally established. However, the mentioned comprehensive meta-analysis recently evaluated SSTR-targeted PRRT using ^90^Y-DOTATOC, ^177^Lu-DOTATOC, ^177^Lu-DOTATATE, or combinations in histologically validated meningioma [49]. The evaluated studies in the meta-analysis applied SSTR-targeted PRRT on treatment-refractory meningioma patients with exhausted conventional treatment modalities such as: surgery, fractioned or stereotactic radiotherapy, or chemotherapy prior to SSTR-targeted therapy. Before the start of therapy, intense SSRT-expression was confirmed in the meningiomas using PET/CT. Independent of the tumor grade for up to six cycles of therapy, a median of 12,590 Mbq (range: 1688–29,772) was applied for treatment. Manageable mild transient hematotoxicity (anemia, leukopenia, lymphocytopenia, and thrombocytopenia) was the major side effect co-occurring under PRRT and as such, constant evaluation of hemoglobin, granulocytes, leukocytes, and thrombocytes is warranted before and between PRRT cycles. Overall, PRRT showed a positive treatment effect with manageable side effects. Treatment response was evaluated using different radiological assessment protocols finding favorable overall survival in the combined outcome of all studies [49].

The meta-analysis underlines the small number of cases that have been published so far. At present, no larger controlled trials have been conducted. One randomized trial is currently recruiting patients with recurrent or progressive meningiomas for treatment with the radiolabeled somatostatin antagonist 177Lu-satoreotide (PROMENADE study, NCT04997317). An overview of the studies already conducted can be seen in Table 1.

## 3. Outlook

### 3.1. Pretreatment PET Imaging vs. Routine SSTR Expression Assessment

Before PRRT can be considered for meningioma treatment, PET imaging is used to provide important information about the expected treatment efficacy. PET imaging with ^68^Ga-labeled somatostatin analogs is used to estimate the potential for radionuclide uptake in meningioma tissue during PRRT. Similarly, this method was demonstrated in an imaging study of 11 meningioma patients who underwent several whole-body and single-photon emission computed tomographies (SPECT/CT) during PRRT to assess the radionuclide kinetics [39]. Furthermore, a correlation between the DOTATATE/-TOC PET imaging signal and the response to PRRT was demonstrated in a retrospective analysis of 20 patients suffering from recurrent or high-grade meningioma. The authors also performed immunohistochemical staining of SSTR2A and observed that a high expression signal was associated with improved 6-month progression-free survival [50].

Only a few studies have investigated the expression of the different somatostatin receptors in meningioma tissue, and different methods were applied in these studies. Our recent retrospective analysis described the distribution of the immunohistochemical expression of all five somatostatin receptors in 726 meningiomas, which is the largest analysis of this kind so far [24]. Whether a certain level of SSTR expression is associated with PRRT efficacy is unclear. Currently, SSTR expression as assessed by PET imaging is used to predict treatment efficacy. However, additional immunohistochemical studies on the distribution of SSTRs, and especially their correlation to DOTATATE/DOMITATE PET data, are still warranted. This would allow the definition of an immunohistochemical scoring cutoff that may have predictive value regarding PRRT. Because immunohistochemical staining is easy to perform and analyze, it could be integrated in the routine meningioma workup and possibly replace the more expensive PET imaging for the pre-PRRT assessment. However, the expression of SSTRs may not be homogeneous throughout the entire tumor tissue. Consequently, the immunohistochemical detection of SSTRs and the calculation of total SSTR expression from intraoperative tumor samples may not be representative of the whole tumor. Moreover, SSTR expression may change over time, and the immunohistochemical results from tumor tissue that was resected several years ago may not be representative of the tissue in a recurring tumor later in the disease course. Therefore, a noninvasive assessment of whole-tumor SSTR expression through PET imaging will probably remain a practical, useful, and reliable alternative, despite its costs. Additionally, more experience with grading SSTR expression must be gained, and a standardized scoring system needs to be established.

### 3.2. Development of New Substances and the Potential of Tailored PRRT

Peptide receptor radionuclide therapy to treat meningiomas is based on its interaction with the somatostatin receptor 2A expressed in meningioma cells. The somatostatin analog applied in PRRT is octreotide, which has a high affinity to SSTR2A [51,52,53]. Patients who suffer from a meningioma that does not show sufficient SSTR expression through PET imaging should theoretically not be considered for PRRT. Our recent retrospective analysis has demonstrated that SSTR1 and SSTR5 are also highly expressed in many meningiomas. SSTR3 and SSTR4, by contrast, show low immunohistochemical expression [24]. However, there is currently very little knowledge about the link between immunohistochemical somatostatin receptor expression and treatment efficacy. A strong immunohistochemical signal does not necessarily mean that the PRRT will be effective. In our opinion, if the indication of this treatment approach is to be widened, it is crucial to explore the targetability of somatostatin receptors other than SSTR2A. An important factor that supports this idea is the variability of the distribution of SSTR expression among clinical subgroups. For example, SSTR2A expression is significantly lower in neurofibromatosis type 2 meningiomas [24]. Therefore, alternative substances to octeotride should be considered as a vehicle for PRRT. For example, pasireotide has a high affinity to somatostatin receptors 1, 2, 3, and 5 [54], making it a potential candidate for a more efficacious PRRT. In addition, it has not yet been shown whether PRRT would be more efficacious if it were directed towards multiple SSTRs. Extending this idea leads to the prospect of applying a patient-tailored mixture of PRRT vehicles designed after a thorough tissue or imaging analysis of the somatostatin receptor profile of an individual meningioma. Routine immunohistochemistry seems to be an ideal tool for this because it is inexpensive and easy to implement. However, standardized scoring is necessary to ensure comparable results. We utilized the scoring system described by Barresi et al., 2008, which incorporates the staining intensity and area of immunopositivity into a product called the intensity distribution score [25]. Computational quantification can potentially eliminate interobserver variance, which is the main limitation of this score.

Another point which could ensure a better efficacy of PRRT is, as already mentioned in the previous chapter, the intra-arterial application of PRRT. Vonken et al. were able to demonstrate a higher tracer accumulation in the tumor tissue [42]. Apart from another case report and a small study [41,43], which also showed a better uptake in meningioma, further studies are missing. It would therefore be desirable to conduct future studies on the effect of intra-arterial application on the difference in efficacy, overall survival and progression-free survival.

### 3.3. Refining PRRT through Preclinical Models

Preclinical research in animal models may also help to develop novel personalized therapies by refining the in vivo evaluation of receptor expression in multiple tumor entities. For example, high SSTR2A affinity to radiolabeled octreotide has been previously characterized in rats [53]. Moreover, Soto-Montenegro et al. successfully demonstrated the feasibility of imaging somatostatin analogs using PET/CT in a mouse meningioma xenograft model [55]. Our literature review found that although several meningioma animal models have been established [56,57], preclinical PRRT evaluations are still extremely limited. Specifically, the preclinical evaluation of receptor subtypes focusing on affinity and distribution in multiple tumor grades is lacking. We believe that this information would be highly relevant to validate novel radiopeptide candidates.

Preclinical research has focused on improving the current SSTR2A analogs. For example, the theranostic treatment efficacy of ^67^Cu-CuSarTATE was shown to be similar to that of ^177^Lu-DOTATATE in a mouse pancreas tumor model, identifying a novel agent for dosimetry calculation in humans [58]. SSTR2 expression has also been demonstrated in tumor models of pheochromocytoma, small cell lung cancer and thyroid cancer in mice [59,60,61]. Recent work has also focused on the evaluation of SSTR2 using ^177^Lu-DOTATATE in combination with small-molecule poly (ADP-ribose) polymerase-1 (PARP) inhibitors, which showed an increased antitumor efficacy in mice [62].

In our opinion, more knowledge is needed about the distribution of SSTRs other than SSTR2A, in order to expand the armamentarium of PRRT. At the same time, more data is required regarding the dynamics of receptor expression in the tumor over time and the influence of adjuvant treatments, such as radiotherapy on SSTR expression. This could be easily accomplished preclinically, where noninvasive in vivo PET imaging can be performed longitudinally and be directly corroborated by immunohistochemistry. This could also provide preliminary evidence on treatment efficacy and the receptor expression profile of specific tumor types. At the same time, these experiments would increase the characterization and evaluation of novel theranostic radiotracers poised for clinical application. 

## 4. Conclusions

Peptide receptor radionuclide therapy is an innovative treatment approach for meningiomas, with a large and untapped potential.

## Figures and Tables

**Table 1 jcm-11-02364-t001:** Studies that have investigated the effect of PRRT on meningiomas.

Reference	Type of Study	Cohort	Response	PFS (Months)	OS (Months)	Adverse Events/Toxicity
Bartolomei et al. [6]	prospective	n = 29	stabilization n = 19 progression n = 10	6 (from end of PRRT)	40	white blood cells n = 18 renal n = 1
Hartrampf et al. [46]	prospective	n = 10	stabilization n = 7 progression n = 3	91.1	105	none
Kreissl et al. [47]	prospective	n = 10	stabilization n = 8 Partial remission n = 1 complete remission n = 1	-	-	none
Kertels et al. [48]	retrospective	n = 11	stabilization n = 6 no response n = 5	12	37	temporary leukopenia n = 53 thrombozytopenia n = 15 renal n = 3 liver n = 1
Seystahl et al. [50]	retrospective	n = 20	stabilization n = 10 progression n = 10	5.4	not reached	lymphocytopenia 70%

PFS: progression-free survival. OS: overall survival. PFS and OS are given as mean values.

## Data Availability

Not applicable.

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
