# Peer review of "The Current Role of Peptide Receptor Radionuclide Therapy in Meningiomas"

_jcm, 2022, doi:10.3390/jcm11092364_

Round 1

Reviewer 1 Report

The topic of this review is extremely interesting. In the light of new trend in teragnostic application, a detailed revision about specific approach to the treatments of meningioma could be extremely useful.

Despite this aim the paper presents some limitations, it is not extremely innovative. The authors have to deepen the chapter about the already existing data in terms of efficacy and safety of PRRT in meningioma. Particularly, they can add some tables to summarize these results and a quick guide to manage this therapy.

Nevertheless is interesting the discussion and commentary about the different receptor expression and potential role of new substances for tailored PRRT. 

The paragraph on preclinical models could be more succinct.

A real discussion is missing.

Author Response

Reviewer 1:

The topic of this review is extremely interesting. In the light of new trend in teragnostic application, a detailed revision about specific approach to the treatments of meningioma could be extremely useful.

Despite this aim the paper presents some limitations, it is not extremely innovative. The authors have to deepen the chapter about the already existing data in terms of efficacy and safety of PRRT in meningioma. Particularly, they can add some tables to summarize these results and a quick guide to manage this therapy.

We agree with the reviewer that there is a definite need to deepen the data on the use of PRRT for meningiomas. Unfortunately, the available studies are limited. Accordingly, to get a better overview we have prepared a table reflecting the existing studies with the most important results. We also gladly accepted the suggestion to add a short guideline on therapy management, describing which patients are eligible for this therapy and on which parameters attention should be paid. We have therefore added the following paragraph to the chapter “Clinical application and experience in meningioma”:

„Explicit guidelines for the treatment of Meningioma using SSTR-targeted PRRT have still yet to be internationally established. However, the mentioned comprehensive meta-analysis recently evaluated SSTR-targeted PRRT using 90Y-DOTATOC, 177Lu-DOTATOC, 177Lu-DOTATATE, or combinations in histologically validated meningioma. The evaluated studies in the meta-analysis applied SSTR-targeted PRRT on treatment-refractory meningioma patients with exhausted conventional treatment modalities such as: surgery, fractioned or stereotactic radiotherapy or chemotherapy prior to SSTR-targeted therapy. Before the start of therapy, intense SSRT-expression was confirmed in the meningiomas using PET/CT. Independent of the tumor grade for up to 6 cycles of therapy, a median of 12,590 Mbq (range: 1,688-29,772) was applied for treatment. Manageable mild transient Hematotoxicity (anemia, leukopenia, lymphocytopenia and thrombocytopenia) was the major side-effect co-occurring under PRRT and as such, constant evaluation of hemoglobin, granulocytes, leukocytes and thrombocytes is warranted before and between PRRT cycles. Overall, PRRT showed a positive treatment effect with manageable side effects. Treatment response was evaluated using different radiological assessment protocols finding favorable overall survival in the combined outcome of all studies.“

Nevertheless is interesting the discussion and commentary about the different receptor expression and potential role of new substances for tailored PRRT. 

The paragraph on preclinical models could be more succinct.

With this chapter, we wanted to highlight again how sparse the current data on somatostatin receptors, or PRRT in meningiomas is. The currently used information for this therapy is based on knowledge gained from a few small studies from different tumor entities. It is difficult to abbreviate this chapter, as we wanted to reiterate that there is a high need for further preclinical studies to optimize therapy with SSTRs and to identify novel compounds for this purpose. We think that all the information mentioned makes an important contribution to the understanding of this.

A real discussion is missing.

We agree with the reviewer that there is no separate chapter to discuss the topic. However, we have tried to accommodate the discussion in the chapter outlook with its three subchapters and to discuss the various aspects directly in each case when they are named.

We thank the reviewer for the valuable comments and appreciate the opportunity to optimize the review.

Reviewer 2 Report

A well written and very nice overview on the current literature on this toppic.

One item in the outlook might deserve some attentenion. There are many examples of intra arterial application of radiopharmaceuticals for diagnostic and therapeutic procedures that show a much higher uptake in the target. Although this strategy was briefly mentioned in the paragraph on clinical applications, it might be worthwhile to explore the effects on OS and PFS in prospective studies as mentioned by Vonken et al ref 43.

Author Response

Reviewer 2:

A well written and very nice overview on the current literature on this toppic.

One item in the outlook might deserve some attentenion. There are many examples of intra arterial application of radiopharmaceuticals for diagnostic and therapeutic procedures that show a much higher uptake in the target. Although this strategy was briefly mentioned in the paragraph on clinical applications, it might be worthwhile to explore the effects on OS and PFS in prospective studies as mentioned by Vonken et al ref 43.

We thank the reviewer for raising this point. It has indeed been shown that there is better uptake of the tracer when intra-arterial administration is used. Unfortunately, there is little or no data on the use and outcomes of progression-free survival or overall survival in meningioma patients. We have added the following text passage in the chapter “Development of new substances and the potential of tailored PRRT”:

„Another point which could ensure a better efficacy of PRRT is, as already mentioned in the previous chapter, the intra-arterial application of PRRT. Vonken et al. were able to demonstrate a higher tracer accumulation in the tumor tissue [43]. Apart from another case report and a small study[42,44], which also showed a better uptake in the meningioma, further studies are missing. It would therefore be desirable to conduct future studies on the effect of intra-arterial application on the difference in efficacy, overall survival and porogression-free survival.“